# Second-Order Side-Channel Analysis Based on Orthogonal Transform Nonlinear Regression

**DOI:** 10.3390/e25030505

**Published:** 2023-03-15

**Authors:** Zijing Jiang, Qun Ding

**Affiliations:** Electronic Engineering College, Heilongjiang University, Harbin 150080, China

**Keywords:** block cipher, FPGA, linear regression, orthogonal transform, template analysis

## Abstract

In recent years, side-channel analysis technology has been one of the greatest threats to information security. SCA decrypts the key information in the encryption device by establishing an appropriate leakage model. As one of many leakage models, the XOR operation leakage proposed by linear regression has typical representative significance in side-channel analysis. However, linear regression may have the problem of irreversibility of a singular matrix in the modeling stage of template analysis and the problem of poor data fit in the template analysis after the cryptographic algorithm is masked. Therefore, this paper proposes a second-order template analysis method based on orthogonal transformation nonlinear regression. The irreversibility of a singular matrix and the inaccuracy of the model are solved by orthogonal transformation and adding a negative direction to the calculation of the regression coefficient matrix. In order to verify the data fitting effect of the constructed template, a comparative experiment of template analysis based on regression, Gaussian, and clustering was carried out on SAKURA-G. The experimental results show that the second-order template analysis based on orthogonal transformation nonlinear regression can complete key recovery without sacrificing the performance of regression estimation. Under the condition of high noise and high order template analysis, the established template has good universality.

## 1. Introduction

Since Kocher proposed the timing analysis method [1] in 1996, side-channel analysis, a unique cryptanalysis method distinct from classical cryptanalysis, has become a research hotspot in the field of cryptography, after more than 20 years of development with its powerful analysis ability and wide application range. General classes of side-channel analysis include timing analysis, power analysis [2,3,4,5,6,7], template analysis [8,9,10,11,12,13,14], electromagnetic analysis [14,15,16], collision attack [17,18], fault analysis [19,20,21], and artificial intelligence side-channel analysis [22,23,24].

Template analysis is a new side-channel attack method proposed by Chari et al. in 2002. This method has received keen attention since it was proposed. Due to the noise interference when collecting the power consumption curve of cryptographic equipment, if the signal-to-noise ratio of the collected power consumption signal is relatively low, the traditional side-channel attack method may be limited, resulting in the failure of the attack. Therefore, the attacker must use other methods to obtain the key. However, template analysis can effectively use the noise in the power consumption information, so the noise will not affect its attack results. Shortly after the template analysis was proposed, scholars proposed a random attack using linear regression in the analysis stage [25]. Subsequently, Wang et al. [26] proposed a ridge regression-based template analysis in 2018. With the deepening of the research on template analysis, researchers realize some weaknesses in regression analysis.

First, there may be the problem of irreversibility caused by singular matrices in linear regression analysis. In order to solve this problem, this paper proposes to project the data of the characteristic matrix from the original space to the new space through orthogonal transformation. At this time, the linearly related variables are transformed into linearly uncorrelated variables, which can solve the problem of irreversibility caused by singular matrices in linear regression analysis without sacrificing the performance of regression estimation.

Secondly, compared with linear regression, ridge regression [26,27,28] improves its data fitting effect, however, because the ridge regression coefficient takes up too much space contributed by the original feature matrix, it may lead to large offset when the cryptographic algorithm is masked, causing it to inaccurately fit the data. To solve this problem, nonlinear regression is proposed in this paper. On the basis of linear regression, a negative direction is added to the calculation of the regression coefficient matrix to make the least squares estimation of the regression coefficients closer to the actual data.

Finally, in order to construct a generalized template for side-channel analysis, a second-order template analysis based on orthogonal transform nonlinear regression model is proposed in this paper. The template constructed by our method has superior universality and has significant advantages over the existing template analysis based on regression in the efficiency of key guessing under high noise and high order conditions.

Section 2 introduces the necessary knowledge of power consumption model, linear regression, and ridge regression. In Section 3, a second-order template analysis based on orthogonal transform nonlinear regression is proposed. In Section 4, the possible causes of the singular matrix are deduced, and the feasibility of the orthogonal transformation method to solve this problem is analyzed. In Section 5, the key recovery efficiency and computational complexity of second-order and higher-order template analysis based on orthogonal transformation nonlinear regression are verified through comparative experiments on SAKURA-G. Finally, we conclude this paper in Section 6.

## 2. Related Works

### 2.1. Power Model

In power analysis, it is generally necessary to establish a corresponding relationship between the operation data of the device and the simulation value of power consumption to characterize the power consumption of cryptographic devices. In power consumption attack, Hamming distance model and Hamming weight model are two principal models [29] to describe circuit power consumption.

Hamming weight refers to the number of 1 in the binary representation of *v*. The Hamming weight model is more suitable for microcontrollers using a precharge bus. When the intermediate value of the cryptographic algorithm is copied from memory to register, or other operations related to the data occur, Hamming weight leakage will occur. This power consumption is usually related to Hamming weight, which is illustrated in (1):(1)T=a·HW(v)+b
where *v* represents the intermediate value of the cryptographic operation, and *HW* represents the Hamming weight. *T* represents power consumption, *a* is the proportion coefficient of power consumption, and *b* represents leakage and noise not related to the processed data.

Hamming distance refers to the Hamming weight after two values are XOR. The Hamming distance model is suitable for register bit reversal in hardware implementation. When the clock arrives, the register bit turns over, and the number of turns is used to describe the power consumption value at that time. This power consumption is usually related to the Hamming weight, as shown in (2):(2)$$T=a\cdot HD(v_{1},v_{2})+b$$
where $v_{1}$ is recorded as the state before the circuit change, $v_{2}$ is recorded as the state after the change, and *HD* represents Hamming distance.

In conclusion, the Hamming distance model is more suitable for describing the power consumption caused by bit inversion in FPGA registers.

### 2.2. Regression Analysis

Regression analysis originates from statistical theory. Regression analysis is a common method to study the logic between variables, make statistical analyses and build models. Regression analysis is commonly used in practice, so it naturally derives a variety of regression analyses. The most famous are linear regression and logistic regression.

Linear regression refers to a linear regression problem in which a sample has multiple characteristics. For a sample *i* with *n* characteristics, its linear regression overall regression model can be given in (3):(3)$$y_{i}=w_{0}+w_{1}x_{i,1}+w_{2}x_{i,2}+\ldots +w_{n}x_{i,n}$$
where $w_{j}$ is called the regression coefficient, $x_{i,j}$ is the different characteristics of sample *i*, and $y_{i}$ is the target variable.

If *m* samples are considered, the matrix form of the regression result is shown in (4):(4)$$\left[\begin{array}{c} {\hat{y}}_{1}\\ {\hat{y}}_{2}\\ {\hat{y}}_{3}\\ \ldots \\ {\hat{y}}_{m} \end{array}\right]=\left[\begin{array}{cccc} 1 & x_{1,1} & \ldots & x_{1,n}\\ 1 & x_{2,1} & \ldots & x_{2,n}\\ 1 & x_{3,1} & \ldots & x_{3,n}\\ \ldots & \ldots & \ldots & \ldots \\ 1 & x_{m,1} & \ldots & x_{m,n} \end{array}\right]\left[\begin{array}{c} w_{0}\\ w_{1}\\ w_{2}\\ \ldots \\ w_{n} \end{array}\right]$$
where *Y* is the column vector containing the regression results of all m samples, *X* is a characteristic matrix with structure (*m*, *n* + 1), and *W* can be regarded as a column matrix with structure (*n* + 1, 1).

In linear regression, the definition of loss function can be expressed as (5):(5)$$\begin{array}{ll} \sum_{i=1}^{m}e_{i}^{2} & =\sum_{i=1}^{m}{\left(y_{i}-{\hat{y}}_{i}\right)}^{2}\\ & =\sum_{i=1}^{m}{\left(y_{i}-X_{i}W\right)}^{2}\\ & =\sum_{i=1}^{m}{\left(y_{i}-w_{0}-w_{1}x_{i,1}-\ldots -w_{n}x_{i,n}\right)}^{2} \end{array}$$

The solution of the least squares normal equation is solved by minimizing the sum of the squares of the residuals between the real value and the predicted value, and the least squares estimate $w_{j}$ of the regression coefficient is obtained.
(6)$$\{\begin{array}{l} \frac{\partial }{\partial w_{1}}\sum_{i=1}^{m}e_{i}^{2}=-2\sum_{i=1}^{m}(y_{i}-w_{0}-w_{1}x_{i,1}\ldots -w_{n}x_{i,n})=0\\ \frac{\partial }{\partial w_{2}}\sum_{i=1}^{m}e_{i}^{2}=-2\sum_{i=1}^{m}(y_{i}-w_{0}-w_{1}x_{i,1}\ldots -w_{n}x_{i,n})x_{i,1}=0\\ \ldots \\ \frac{\partial }{\partial w_{n}}\sum_{i=1}^{m}e_{i}^{2}=-2\sum_{i=1}^{m}(y_{i}-w_{0}-w_{1}x_{i,1}\ldots -w_{n}x_{i,n})x_{i,n}=0 \end{array}$$

According to (6), it can be seen that the residual vector e is orthogonal to each characteristic of sample *i*. The least squares estimation $\hat{W}$ of linear regression coefficients is shown in (7):(7)$$\hat{W}={\left(X^{T}X\right)}^{-1}X^{T}Y$$

According to (7), it can be seen that the regression coefficient $\hat{W}$ will be obtained by *Y* and *X* together.

### 2.3. Template Analysis Based on Linear Regression

Template analysis is generally divided into two processes: modeling phase and template matching phase, as shown in Figure 1. In the modeling phase, linear regression establishes a mathematical relationship model between the power consumption and each bit of the key based on the Hamming weight or Hamming distance leakage model, and then directly uses this model to match the actual power consumption in the template matching phase to guess the correct key.


**Modeling stage**


The main purpose of the modeling stage is to characterize the device leakage through the power consumption curve of a device with a known key and find the power consumption model.

The linear regression can be matched to the power model of (1) and (2). Hamming distance model is more suitable for describing the power consumption caused by bit inversion in FPGA registers. Therefore, this paper takes the Hamming distance model as an example for theoretical deduction, as shown in (8):(8)$$\begin{array}{ll} T & =a\cdot HD(v_{1},v_{2})+b\\ & =a\cdot HW(v_{1}⊕v_{2})+b\\ & =a\cdot \sum_{i=1}^{n}\left(v_{1}[j]⊕v_{2}[j]\right)+b \end{array}$$

The relationship matrix between m power traces and the intermediate value can be obtained by linear regression method, as shown in (9):(9)$$\left[\begin{array}{c} T_{1}\\ T_{2}\\ \ldots \\ T_{m} \end{array}\right]=\left[\begin{array}{cccc} 1 & v_{1,1}[1]⊕v_{1,2}[1] & \ldots & v_{1,1}[n]⊕v_{1,2}[n]\\ 1 & v_{2,1}[1]⊕v_{2,2}[1] & \ldots & v_{2,1}[n]⊕v_{2,2}[n]\\ 1 & \ldots & \ldots & \ldots \\ 1 & v_{m,1}[1]⊕v_{m,2}[1] & \ldots & v_{m,1}[n]⊕v_{m,2}[n] \end{array}\right]\left[\begin{array}{c} a_{0}\\ a_{1}\\ \ldots \\ a_{n} \end{array}\right]$$

A set of regression coefficients A is obtained according to (7) and (9), and the modeling is completed at this time.


**Template matching stage**


The main purpose of the template matching phase is to recover the unknown key in the tested device by using the constructed template.

In the template matching phase, the constructed template is used to calculate the estimated power of all possible key values. The correct key with the best correlation is matched by calculating the correlation coefficient of the estimated power consumption $\hat{Y}$ and the actual power consumption *Y*. The correlation coefficient is calculated as shown in (10):(10)$$\rho =\frac{E(\hat{Y},Y)-E(\hat{Y})E(Y)}{\sqrt{D(\hat{Y})D(Y)}}$$
where *E* represents the expectation, *D* represents the variance, and *ρ* represents the correlation coefficient.

The guess key with the highest correlation coefficient is the correct key.

### 2.4. AES-128 Algorithm

This paper aims to solve the second-order template analysis problem of the AES-128 algorithm with a mask. The AES-128 algorithm is shown in Figure 2. The AES-128 algorithm consists of 10 round functions. Before the round function, XOR is performed between the whitening key and the plaintext. Round functions from round 1 to round 9 include four operations: SubByte, ShiftRow, MixColumn, and AddRounKey. The last round does not perform MixColumn. The following four operation stages of the round function in AES are introduced, respectively. These four operation processes fully confuse the input bits.

**SubByte:** SubByte is actually a simple table lookup operation. The elements in the state matrix are mapped to a new byte by taking the upper 4 bits of the byte as row values and the lower 4 bits as column values.

**ShiftRow:** ShiftRow is a simple left rotation operation. When the key length is 128 bits, row 0 of the state matrix is shifted 0 bytes to the left, row 1 is shifted 1 byte to the left, row 2 is shifted 2 bytes to the left, and row 3 is shifted 3 bytes to the left.

**MixColumn:** MixColumn transformation is realized by matrix multiplication. The state matrix after row shift is multiplied with the fixed matrix to obtain the confused state matrix.

**AddRounKey:** AddRounKey is a bit-by-bit XOR operation between the 128-bit round key and the data in the state matrix.

As shown in Figure 3, the power consumption curve and leakage analysis curve of the complete AES-128 algorithm are shown. The red curve represents the power consumption, and the blue curve represents the leakage analysis. Each peak of the power consumption curve represents a round of encryption (decryption), and the peak position of the leakage analysis curve represents the leakage area. The peak position of the leakage analysis curve corresponds to the eleventh peaks of the power consumption curve, indicating that power leakage occurred in the tenth round of encryption (decryption). Therefore, register A of the AES-128 algorithm in Figure 2 is selected to discuss the efficiency of template analysis.

## 3. Orthogonal Transformation Nonlinear Regression Analysis

### 3.1. Orthogonal Transformation Nonlinear Regression Model

In the second section, the solving principle of linear regression using the least square method is derived, and the regression coefficient *W* is obtained. It can be seen from Equation (7) that there is an inverse operation in the equation. If $X^{T}X$ is a singular matrix, we will not be able to obtain its inverse. This situation exists in the actual power consumption acquisition process. If $X^{T}X$ is a singular matrix, then only a new set of power traces can be collected for analysis. In 2018, Wang et al. [26] proposed a ridge regression method to solve this problem. Coefficient α avoids the influence of a singular matrix. However, in the case where the cryptographic algorithm is masked, because coefficient *α* occupies too much space contributed by the original characteristic matrix in *W*, it may lead to large deviation in *W* and incorrectly fit the real face of the data.

The inability to solve by least squares and poor data fit of existing regressions in the case of masked cryptographic algorithms, can be solved with a new regression model. Therefore, an orthogonal transform nonlinear regression model is proposed in this paper. The orthogonal transformation nonlinear regression model can solve the problem of a singular matrix being irreversible in multivariate linear regression analysis, and has a good data fitting effect in the cases of high noise and high order. The specific implementation steps are as follows:

**Step 1:** Collect the power consumption waveform to form the sample matrix *X*. The m power consumption waveforms of the circuit are collected through the power consumption acquisition platform. Each power consumption waveform has n sampling points, forming an [m, n] sample matrix *X*.

**Step 2:** Calculate the mean value of the sample. Take the average value of each column of the sample matrix *X* to obtain the sample mean vector $\bar{X}$, and its calculation formula is shown in (11):(11)$$\bar{X}=[\begin{array}{cccc} \frac{1}{m}\sum_{i=1}^{m}x_{i,1} & \frac{1}{m}\sum_{i=1}^{m}x_{i,2} & \ldots & \frac{1}{m}\sum_{i=1}^{m}x_{i,n} \end{array}]$$

**Step 3:** Centralized sample matrix. The centralized sample matrix is to subtract the sample mean $\bar{X}$ from each column of sample matrix *X* to obtain the centralized sample matrix $\hat{X}$. The calculation formula of matrix $\hat{X}$ is shown in (12):(12)$$\hat{X}=\left[\begin{array}{cccc} x_{1,1} & x_{1,2} & \ldots & x_{1,n}\\ x_{2,1} & x_{2,2} & \ldots & x_{2,n}\\ \ldots & \ldots & \ldots & \ldots \\ x_{m,1} & x_{m,2} & \ldots & x_{m,n} \end{array}\right]-{\left[\begin{array}{c} {\bar{x}}_{1}\\ {\bar{x}}_{2}\\ \ldots \\ {\bar{x}}_{n} \end{array}\right]}^{T}$$

**Step 4:** Calculate the covariance matrix *C*. Calculate the covariance matrix *C* of the centralized sample matrix $\hat{X}$. The calculation method of [*n*, *n*] matrix *C* is shown in (13):(13)$$C=\frac{1}{m-1}{\hat{X}}^{T}\hat{X}$$

**Step 5:** Calculate the eigenvector matrix and eigenvalue matrix. The covariance matrix *C* is decomposed into eigenvalues, and its eigenvalues and eigenvectors are obtained. The calculation method is shown in (14):(14)$$A^{T}CA=\lambda$$
where *A* is called the eigenvector matrix and *λ* is the eigenvalue diagonal matrix. 

**Step 6:** Orthogonal transformation. The centralized sample matrix $\hat{X}$ is projected into the new space to obtain the matrix $\tilde{X}$ after orthogonal transformation, as shown in (15):(15)$$\tilde{X}=\hat{X}\cdot A_{n\times k}$$

The sample matrix *X*, which may lead to the appearance of a singular matrix in $X^{T}X$, is orthogonally transformed into a new matrix $\tilde{X}$.

**Step 7:** Nonlinear regression. The reconstructed uncorrelated matrix $\tilde{X}$ can only solve the problem that the original singular matrix is irreversible. If we use the least square method to solve the regression coefficient matrix *W* with the reconstructed uncorrelated matrix $\tilde{X}$ according to the linear regression model, the data fitting effect will not be improved. Therefore, in order to obtain a regression model with a better fitting effect in noisy environments, this paper introduces *αW* into the loss function of linear regression, and the definition of the loss function of the improved nonlinear regression model can be expressed as (16):(16)$$\begin{array}{ll} \sum_{i=1}^{m}e_{i}^{2}+\alpha W & =\sum_{i=1}^{m}{\left(y_{i}-{\hat{y}}_{i}\right)}^{2}+\alpha W\\ & =\sum_{i=1}^{m}{\left(y_{i}-X_{i}W\right)}^{2}+\alpha W \end{array}$$

The nonlinear regression model still uses the least square method to obtain the regression coefficient matrix *W*:(17)$$W={\left(X^{T}X\right)}^{-1}(X^{T}Y-AI)$$
where matrix *I* is a unit matrix. The coefficient *A* takes any value, and it is linear regression when *A* = 0.

By increasing the coefficient *A*, a negative direction can be added to the calculation of *W*, so as to limit the size of *W* in parameter estimation and prevent the problem of model inaccuracy caused by too large of a parameter estimation.

### 3.2. Second-Order Template Analysis Based on Orthogonal Transform Nonlinear Regression

Second-order template analysis based on orthogonal transformation nonlinear regression adopts the idea of “divide and conquer” to model, as it calculates the corresponding intermediate value through the known key, reconstructs the intermediate value matrix into a linear uncorrelated matrix, and uses the corresponding regression coefficient matrix to construct the template. Through the constructed template, the correlation coefficients of the estimated power matrix and the actual power matrix of all possible key values are calculated, and the correct key with the optimal correlation is matched. The specific process of second-order template analysis based on orthogonal transform nonlinear regression is given below, as shown in Algorithm 1.


**Modeling stage**


**Step 1:** Collect n modeling power traces *T* of random plaintext with the known key and combine the power consumption points with the centralized multiplication combination function, as shown in (18), where each power traces corresponds to m sampling points. The *j*-th sampling point of the *i*-th power traces is expressed as $T_{i,j}$. Record the corresponding plaintext *P* and ciphertext *C*. Collect a set of matching power traces $T^{\prime }$ of random plaintext with unknown key. Record the corresponding plaintext $P^{\prime }$ and ciphertext $C^{\prime }$.
(18)$${T^{\prime }}_{i}={\left(T_{i,j}-mean(T_{j})\right)}^{2}$$

**Step 2:** According to plaintext *P* and ciphertext *C*, the Hamming distance matrix of the middle value of the known key is calculated as the characteristic matrix $X^{*}$ of regression analysis.

**Step 3:** The Hamming distance matrix $X^{*}$ is reconstructed into a linearly uncorrelated matrix ${\tilde{X}}^{*}$ by orthogonal transformation.

**Step 4:** Using the least square method, the reconstructed uncorrelated matrix ${\tilde{X}}^{*}$ is used as the characteristic matrix. Our proposal solves the regression coefficient matrix $W^{*}$ according to Formula (17), and thus completes the template orthogonal transformation nonlinear regression template construction stage.


**Template matching stage**


**Step 5:** According to plaintext $P^{\prime }$ and ciphertext $C^{\prime }$, calculate the intermediate Hamming distance matrix under the possible values of all unknown keys as the characteristic matrix *X* of regression analysis.

**Step 6:** The linear uncorrelated matrix $\tilde{X}$ is reconstructed by using the eigenvector matrix of the covariance matrix *C* of the modeling stage matrix $X^{*}$, as shown in (15).

**Step 7:** The regression coefficient matrix $W^{*}$ and matrix $\tilde{X}$ are returned to (4) to calculate the estimated power consumption matrix $\hat{Y}$.

**Step 8:** According to (10), the correlation coefficient between the estimated power consumption matrix $\hat{Y}$ and the actual power consumption matrix $Y^{\prime }$ processed by the centralized multiplication combination function is calculated.

The guess key with the highest correlation coefficient is the correct key.
**Algorithm 1** Second-order Template analysis based on orthogonal transform nonlinear regressionBegin for *S* = 1:16/*Template building stage*/  for *p* = 1:*num*   $HD(v^{*}{}_{1},v^{*}{}_{2})\to X^{*}(p,1:8)$
  end  ${\hat{X}}^{*}=X^{*}-{\bar{X}}^{*}$  $C={\hat{X}}^{*}{}^{T}{\hat{X}}^{*}/m-1$  $A^{*}{}^{T}CA^{*}=\lambda^{*}$ ${\tilde{X}}^{*}(1:num,2:9)={\hat{X}}^{*}\cdot A^{*}{}_{n\times k}$ ${\tilde{X}}^{*}(1:num,1)=zeros(num,1)+1$ $Y^{*}={\left(t_{i,j}-mean(t_{j})\right)}^{2}$ $W^{*}={\left({\tilde{X}}^{{*}^{T}}{\tilde{X}}^{*}\right)}^{-1}({\tilde{X}}^{{*}^{T}}Y^{*}-AI)$/*template analysis stage*/ for *k* = 0:255  for *p* = 1:*num*   $HD(v_{1},v_{2})\to X(p,1:8)$  end  $\hat{X}=X-\bar{X}$ $\tilde{X}(1:num,2:9)=\hat{X}\cdot A^{*}{}_{n\times k}$ $\tilde{X}(1:num,1)=zeros(num,1)+1$ $Y={\left(t\_test_{i,j}-mean(t\_test_{j})\right)}^{2}$ $\hat{Y}=\tilde{X}\cdot W^{*}$ for *i* = 1:Leakage_point   Corr(*k* + 1, *i*) = |*ρ*($\hat{Y}$(:,i), Y(:,i))| end   end   [*m*, *n*] = find(corr = max (max(corr)))   Correct_key(1,*S*) = *m* − 1end

### 3.3. Parameter Selection

Before implementing orthogonal transformation nonlinear regression modeling, the attacker should first find an optimal parameter value A. Therefore, in this section, the parameter A in the orthogonal transformation nonlinear regression is selected. Modeling and matching are carried out respectively under the condition that the noise standard deviations are 2 and 4. The parameter set to be selected is a = {0.1, 1, 10, 50, 200, 500, 1000, 2000, 5000, 10,000}. Next, 100 key-guessing experiments are carried out and the average number of key bytes guessed is calculated. 

As shown in Figure 4, for modeling and matching under different noise standard deviations, the optimal solution of the parameter *A* is also different. When *A* ≥ 2000, the number of bytes recovered from the key tends to be stable. Therefore, in the subsequent experiment, we select *A* = 2000 for template construction.

## 4. Theoretical Analysis

In this section, we theoretically study the problems of linear regression analysis. Firstly, we find the cause of paralysis of linear regression analysis by analyzing the principle of the linear regression model. Then, aiming at specific problems, we solve the loopholes of linear regression through orthogonal transformation nonlinear regression.

According to the previous section, the reason for the unavailability of linear regression analysis is that matrix $X^{T}X$ is a singular matrix and irreversible. Next, we analyze the conditions for matrix $X^{T}X$ to become a singular matrix, as shown in (19) below:(19)$$X^{T}X=\left[\begin{array}{ccccc} m & x_{1,1}+x_{2,1}+\ldots +x_{m,1} & x_{1,2}+x_{2,2}+\ldots +x_{m,2} & \ldots & x_{1,n}+x_{2,n}+\ldots +x_{m,n}\\ x_{1,1}+x_{2,1}+\ldots +x_{m,1} & x_{1,1}^{2}+x_{2,1}^{2}+\ldots +x_{m,1}^{2} & x_{1,1}x_{1,2}+x_{2,1}x_{2,2}+\ldots +x_{m,1}x_{m,2} & \ldots & x_{1,1}x_{1,n}+x_{2,1}x_{2,n}+\ldots +x_{m,1}x_{m,n}\\ x_{1,2}+x_{2,2}+\ldots +x_{m,2} & x_{1,2}x_{1,1}+x_{2,2}x_{2,1}+\ldots +x_{m,2}x_{m,1} & x_{1,2}^{2}+x_{2,2}^{2}+\ldots +x_{m,2}^{2} & \ldots & x_{1,2}x_{1,n}+x_{2,2}x_{2,n}+\ldots +x_{m,2}x_{m,n}\\ \ldots & \ldots & \ldots & \ldots & \ldots \\ x_{1,n}+x_{2,n}+\ldots +x_{m,n} & x_{1,n}x_{1,1}+x_{2,n}x_{2,1}+\ldots +x_{m,n}x_{m,1} & x_{1,n}x_{1,2}+x_{2,n}x_{2,2}+\ldots +x_{m,n}x_{m,2} & \ldots & x_{1,n}^{2}+x_{2,n}^{2}+\ldots +x_{m,n}^{2} \end{array}\right]$$

If the matrix $X^{T}X$ is a singular matrix, there must be a linear correlation between two row vectors of matrix $X^{T}X$, and the row vectors can be eliminated by elementary transformation, resulting in the following two cases of (20) and (21):(20)$$\{\begin{array}{c} x_{1,i}+x_{2,i}+\ldots +x_{m,i}=a\cdot m\\ x_{1,i}x_{1,1}+x_{2,i}x_{2,1}+\ldots +x_{m,i}x_{m,1}=a\cdot (x_{1,1}+x_{2,1}+\ldots +x_{m,1})\\ x_{1,i}x_{1,2}+x_{2,i}x_{2,2}+\ldots +x_{m,i}x_{m,2}=a\cdot (x_{1,2}+x_{2,2}+\ldots +x_{m,2})\\ \ldots \\ x_{1,i}x_{1,n}+x_{2,i}x_{2,n}+\ldots +x_{m,i}x_{m,n}=a\cdot (x_{1,n}+x_{2,n}+\ldots +x_{m,n}) \end{array}\begin{array}{cc} & i=1,2,\ldots ,n \end{array}$$
(21)$$\{\begin{array}{c} x_{1,i}+x_{2,i}+\ldots +x_{m,i}=a\cdot (x_{1,j}+x_{2,j}+\ldots +x_{m,j})\\ x_{1,i}x_{1,1}+x_{2,i}x_{2,1}+\ldots +x_{m,i}x_{m,1}=a\cdot (x_{1,j}x_{1,1}+x_{2,j}x_{2,1}+\ldots +x_{m,j}x_{m,1})\\ x_{1,i}x_{1,2}+x_{2,i}x_{2,2}+\ldots +x_{m,i}x_{m,2}=a\cdot (x_{1,j}x_{1,2}+x_{2,j}x_{2,2}+\ldots +x_{m,j}x_{m,2})\\ \ldots \\ x_{1,i}x_{1,n}+x_{2,i}x_{2,n}+\ldots +x_{m,i}x_{m,n}=a\cdot (x_{1,j}x_{1,n}+x_{2,j}x_{2,n}+\ldots +x_{m,j}x_{m,n}) \end{array}\begin{array}{cc} & i,j=1,2,\ldots ,n \end{array}$$

The different characteristic $x_{i,j}$ of the characteristic matrix *X* is composed of Hamming distance of the intermediate value, so the characteristic $x_{i,j}$ can only be taken as 0 and 1, as shown in Figure 5. If any column vector of characteristic matrix *X* is all 0 or all 1, $X^{T}X$ will be an irreversible matrix. 

The orthogonal transformation nonlinear regression proposed in this paper projects the original data of characteristic matrix *X* from the original space to the new space through the eigenvector of the covariance matrix. The value range of characteristic $x_{i,j}\in R$ in the new space is no longer limited to 0 and 1, as shown in Figure 6. Compared with the original matrix space with only 0 and 1, the new matrix space is more complex and changeable, and it is more difficult to have the problem of matrix irreversibility.

### 4.1. Autocorrelation Test

Autocorrelation is mainly to test the correlation degree between the binary sequence to be tested and the new sequence obtained by moving the sequence by *k* bits. The detection method is generally realized by calculating the autocorrelation function of the sequence. For the feature vector in linear regression, it should have very low linear correlation with the new vector after moving any bit. The mathematical expression of its autocorrelation function is defined in (22):(22)$$r_{x}(k)=\underset{N\to \infty }{\lim}\frac{1}{N}\sum_{n=0}^{N-1}x(n)x(n+k)$$

There are two estimation formulas of autocorrelation, namely unbiased estimation and biased estimation. Unbiased estimation can be defined as shown in (23):(23)$$R_{x}(k)=\frac{1}{N-\left|k\right|}\sum_{n=0}^{N-1-\left|k\right|}x_{N}(n)x_{N}(n+k)$$

Mathematical deviation is defined as shown in Equation (24):(24)$$R_{x}(k)=\frac{1}{N}\sum_{n=0}^{N-1-\left|k\right|}x_{N}(n)x_{N}(n+k)$$

Based on the above theoretical analysis, the eigenvectors of the three methods can be tested by autocorrelation. Figure 7a shows the autocorrelation test results of characteristic matrix *X* when the matrix $X^{T}X$ is a nonsingular matrix. It can be seen that the autocorrelation of the eight groups of eigenvectors is very weak, in micro correlation. Figure 7b–d shows the autocorrelation test results of characteristic matrix *X* when the matrix $X^{T}X$ is a singular matrix. The results in Figure 7b,c show that there is a group of eigenvectors with strong autocorrelation in both linear regression and ridge regression, and the autocorrelation estimation *R*(*k*) of other eigenvectors is concentrated at about 0.3. Since the orthogonal transformation nonlinear regression projects the original data of the characteristic matrix *X* from the original space to the new space, the results in Figure 7d show that there are no eigenvectors with strong autocorrelation in the new space, and all eigenvectors have autocorrelation estimation *R*(*k*) closer to 0, showing good randomness.

### 4.2. Cross Correlation Test

In Figure 8, the correlation coefficients between vectors of a matrix $X^{T}X$ of linear regression, and orthogonal transformation regression as well as matrix $X^{T}X+\lambda I$ of ridge regression are obtained, respectively. The correlation coefficient matrix is shown in Figure 8, and yellow indicates complete correlation. As shown in Figure 8a, when the matrix $X^{T}X$ is a nonsingular matrix, each vector is only completely related to its own vector. As shown in Figure 8b, when the matrix $X^{T}X$ is a singular matrix, vector 2 of linear regression is completely related to vector 1. As shown in Figure 8c, when matrix $X^{T}X$ is a singular matrix, although ridge regression solves the problem of accurate correlation to a certain extent, there is still a strong correlation between vector 2 and vector 1, and the correlation coefficient reaches 0.99999984. As shown in Figure 8d, the minimum correlation coefficient between the vectors of the matrix $X^{T}X$ in our proposed orthogonal transformation regression is close to 0, indicating that the matrix $X^{T}X$ does not have collinearity. To summarize, orthogonal transformation regression solves the problem that the original linear regression singular matrix cannot be inversed.

## 5. Security Analysis

In this paper we explore the key guessing efficiency of the orthogonal transformation linear regression and orthogonal transformation nonlinear regression models proposed. This section is compared with linear regression [25], ridge regression [26], Gaussian modeling [13], and cluster-based modeling method [30]. In this section, a power consumption acquisition platform is built based on FPGA to perform key guessing experiments on six template analyses. The attack target is the 10th round function register of the AES-128 algorithm. In the power waveform simulation, it is assumed that the simulation waveform follows the Hamming distance model, the noise follows the Gaussian distribution, and the noise standard deviations are 2 and 4, respectively. The power consumption acquisition platform is shown in Figure 9:

### 5.1. Correlation of Second-Order Template Based on Regression

In order to verify the matching degree between the second-order template constructed by the four regression methods and the actual power consumption, this section uses the correct key to conduct correlation experiments in the leakage area of the tenth-round function of the AES-128 algorithm. It can be seen from Figure 10 that the correlation between the second-order template constructed by linear regression and orthogonal transformation linear regression in the leakage area of the 10th round function of AES-128 algorithm is completely consistent with the actual power consumption, and the maximum Pearson correlation coefficient is 0.052178875016667. The second-order template constructed by ridge regression has the weakest correlation with actual power consumption, and its maximum Pearson correlation coefficient is 0.0464993727243824. The second-order template constructed by orthogonal transformation nonlinear regression has the strongest correlation with actual power consumption, and its maximum Pearson correlation coefficient is 0.0528086329655253. Therefore, in the template matching phase, the second-order template constructed based on orthogonal transformation nonlinear regression makes it easier to separate the correct key from the wrong key.

### 5.2. Calculation Complexity

Modeling under the condition that the noise standard deviation is 2, 100 experiments are conducted for the four methods under a different number of template matching waveforms, and the comparison of the average calculation complexity of the 100 experiments is shown in Figure 11. It can be seen from Figure 11 that the computational complexity of the template analysis based on linear regression and template analysis based on ridge regression is almost the same. The template analysis based on orthogonal transformation regression proposed in this paper increases the amount of computation due to the projection of the characteristic matrix from the original space to the new space, so the computational cost is slightly higher than that of linear regression and ridge regression, but this computational cost is insignificant compared with the improvement of the attack efficiency in the template matching.

### 5.3. Guessing Entropy Experiment of Second-Order Template Based on Regression

We verify the universality of the template constructed by second-order template analysis based on orthogonal transformation nonlinear regression. In the case of collecting different numbers of modeling curves and template-matching curves, 100 groups of key-guessing experiments were conducted, in which the measurement unit is guessing entropy. Guessing entropy is the average ranking of 100 groups of correct keys. The set of the number of selected modeling curves is [20, 100, 200, 500, 700, 1000]. The number range of template matching curves is [1000, 10,000], and the step size is 1000.

The guessing entropy of the four methods under different number of modeling curves and template-matching curves is shown in Figure 12. It can be seen from Figure 12 that the guessing entropy of the four methods gradually approaches 1 as the number of template-matching curves increases. In addition, regression models can build effective templates with few modeling curves. Comparing Figure 12a,c, it can be seen that although orthogonal transformation can solve the irreversibility of a singular matrix, orthogonal transformation linear regression has no effect on key guessing. The two methods have exactly the same guessing entropy results under a different number of modeling curves and template-matching curves. Although ridge regression has advantages in convergence speed compared with orthogonal transformation linear regression and linear regression, only the guessing entropy of template analysis based on orthogonal transformation nonlinear regression reaches 1 and the convergence speed is the fastest among the four methods under the condition of a limited number of template-matching curves. This means that the template constructed by the orthogonal transformation nonlinear model is more effective for the AES-128 algorithm with first-order mask protection.

### 5.4. Guessing Entropy Experiment of Second-Order Template Analysis under Different Noise Conditions

Since noise will affect the efficiency of the template analysis, this section will discuss the efficiency of key guessing by modeling and template matching under different noise conditions. Since the guessing entropy of the orthogonal transformation linear regression model and the linear regression model is completely consistent in the previous section, this section only uses the other three regression models for key guess experiments and adds the Gaussian modeling method and cluster-based modeling method. The study has carried out 100 experiments under 20 modeling curves and the different number of template-matching curves, and still uses guessing entropy to measure the efficiency of key guessing.

It can be seen from Figure 13 and Figure 14 that when modeling under the condition that the standard deviation of noise is 2, the guess entropy curves of ridge regression and linear regression are intertwined up and down, and the guessing entropy gradually approaches 1 as the template-matching curve increases. In this scenario, the nonlinear regression based on orthogonal transformation only needs 7000 and 80,000 power traces, and the guessing entropy reaches 1, which means that all bytes of the guess key are successfully recovered. Compared with ridge regression and linear regression, our method reduces the power traces required for key recovery by at least 30% and 20%, respectively. Gaussian modeling method and cluster-based modeling method have a high guess entropy, which is not enough to recover the correct key under the limited number of power traces.

To verify the modeling effect of the regression model under noisy conditions, this section conducts modeling under the condition that the noise standard deviation is 4. It can be seen from Figure 15 and Figure 16 that since ridge regression coefficients occupy too much space contributed by the original characteristic matrix, the convergence rate of ridge regression guessing entropy is even lower than that of linear regression. Under the condition of limited template matching curve, the guessing entropy of linear regression and ridge regression does not reach 1. At this time, nonlinear regression based on orthogonal transformation only needs 8000 and 10,000 power traces, and its guessing entropy reaches 1 to complete key convergence. By applying noise in the modeling stage and template-matching stage, guess entropy shows that the Gaussian modeling method and cluster-based modeling method are still insufficient to recover the correct key. The experimental results show that the nonlinear regression based on orthogonal transformation proposed in this paper has the optimal key guess efficiency whether modeling or template matching under various noise scenarios.

### 5.5. Guessing Entropy Experiment of Higher-Order Template Analysis 

In order to verify the modeling effect of the high-order model of orthogonal transformation nonlinear regression, this section constructs second-order, fourth-order, and eighth-order templates based on five methods for key-guessing experiments, and also uses guessing entropy to measure the efficiency of key guessing under the condition of 20, 30,000 and 150,000 modeling curves.

It can be seen from Figure 17, Figure 18 and Figure 19 that the template-matching curve required for guessing the key of the template constructed by the five methods increases exponentially with the increase of the order. The guessing entropy curve of the second-order template constructed by ridge regression and linear regression is intertwined, but with the increase of the order, because the ridge regression coefficient occupies too much space contributed by the original characteristic matrix, the guessing entropy of the fourth-order templates is even higher than that of the linear regression. In the scenario of the fourth-order template, the guessing entropy of nonlinear regression based on orthogonal transformation proposed in this paper only needs 40,000 template curves to reach 1, while the linear regression needs 80,000, and the guessing entropy of 1.3125 does not converge to 1 when the ridge regression is 100,000. The guess entropy of linear regression and nonlinear regression based on orthogonal transformation is completely consistent at the eighth-order template. Although ridge regression converges quickly when the template curve is insufficient, the guessing entropy of linear regression and nonlinear regression based on orthogonal transformation converges to 1 at 120,000 template curves. At this time, the guessing entropy of ridge regression is 1.0625, until the guessing entropy of 140,000 template curves ridge regression completely converges. The Gaussian modeling method and cluster-based modeling method have a worse effect on guessing the key in higher-order template analysis. The guessing entropy convergence speed of nonlinear regression based on orthogonal transformation, whether in order 2, 4, or 8, with limited number of template-matching curves, reaches 1 at the fastest time to complete the key convergence, indicating that the higher-order modeling effect of nonlinear regression based on orthogonal transformation is optimal and stable.

## 6. Conclusions

This paper discusses the irreversibility of the singular matrix in the high-dimensional feature space at the template-construction stage in the linear regression template analysis, and the data fitting effect of the regression model when the cryptographic algorithm is masked. In this paper, a second-order template analysis method based on orthogonal transformation nonlinear regression is proposed. Our method uses an orthogonal transformation to convert linear correlated variables into linear uncorrelated variables, which can solve the irreversibility problem of singular matrix in linear regression analysis without sacrificing the performance of regression estimation. In order to verify the data fitting of the template constructed by orthogonal transformation nonlinear regression, the key recovery efficiency and computational complexity under different noise conditions, different order, and the different number of modeling curves and template matching curves are compared on SAKURA-G. Experimental results show that the scheme has good universality and key guessing efficiency in high-noise and high-order template analyses.

## Figures and Tables

**Figure 1 entropy-25-00505-f001:**
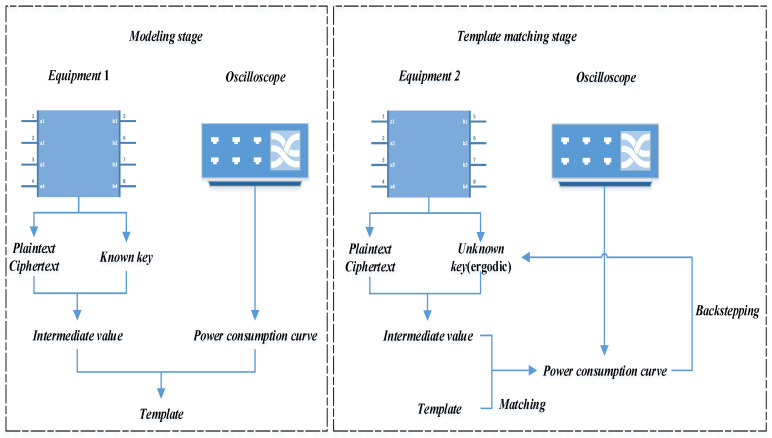
Template analysis.

**Figure 2 entropy-25-00505-f002:**
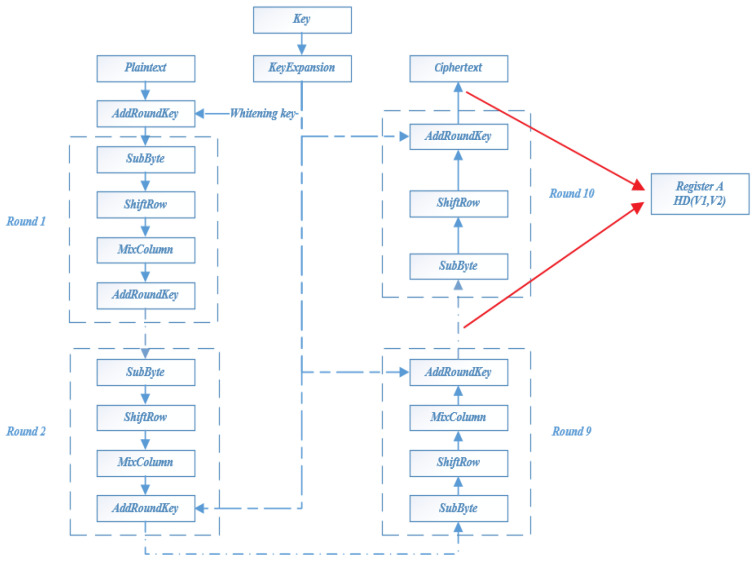
AES-128 algorithm.

**Figure 3 entropy-25-00505-f003:**
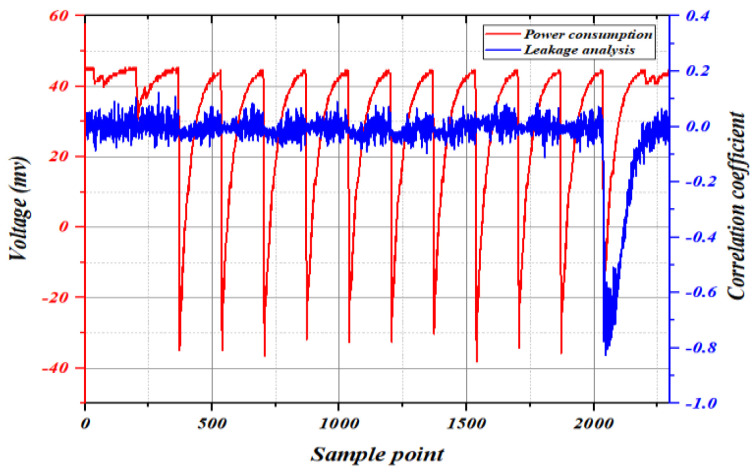
AES-128 leakage analysis based on FPGA.

**Figure 4 entropy-25-00505-f004:**
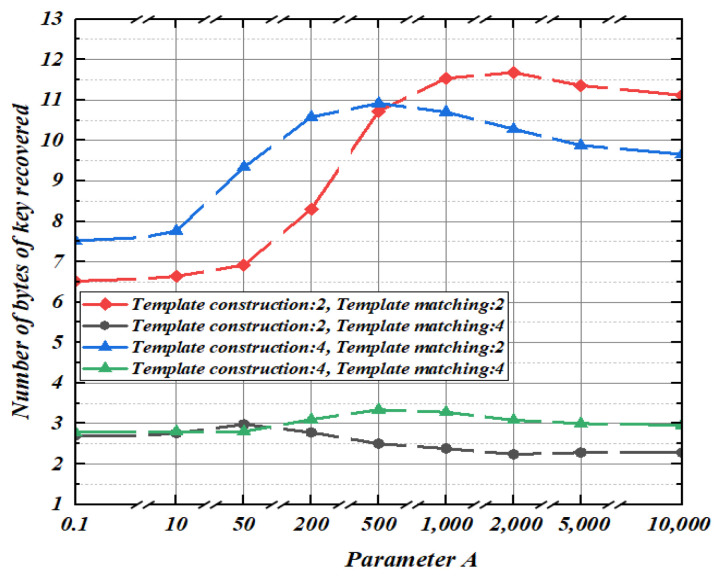
Average bytes of key recovery under different parameter values.

**Figure 5 entropy-25-00505-f005:**
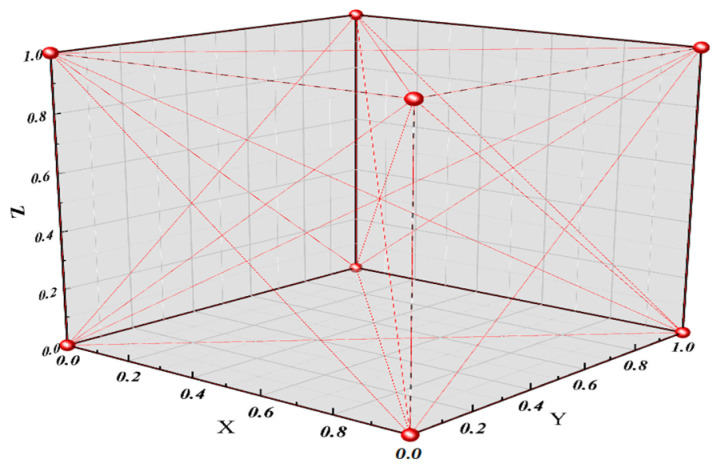
Original matrix space.

**Figure 6 entropy-25-00505-f006:**
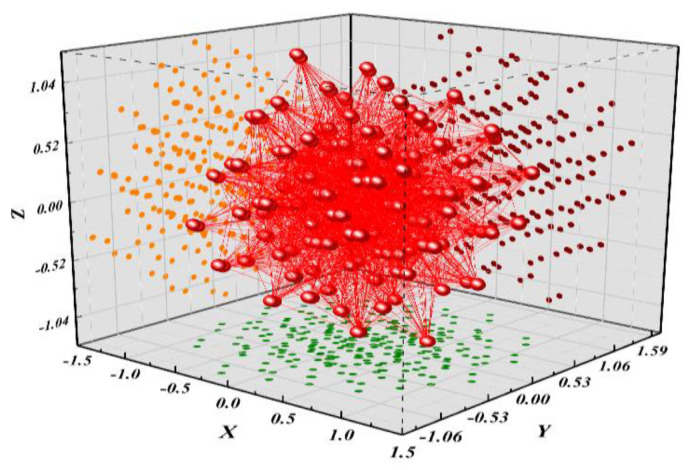
Orthogonal transformation matrix space.

**Figure 7 entropy-25-00505-f007:**
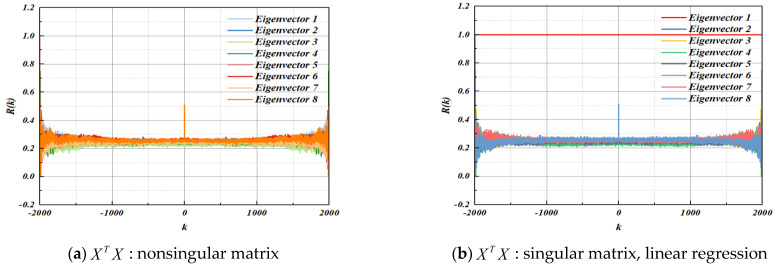
Characteristic matrix X autocorrelation test.

**Figure 8 entropy-25-00505-f008:**
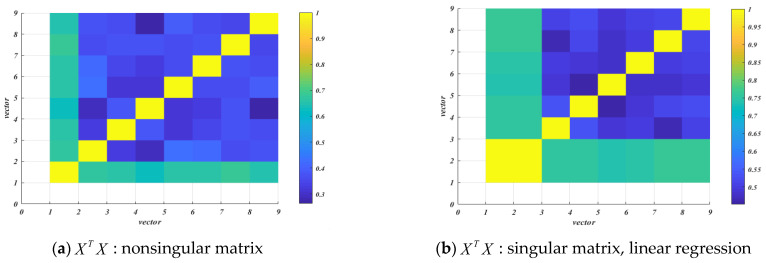
Cross correlation test of vectors of matrix $X^{T}X$ and $X^{T}X+\lambda I$.

**Figure 9 entropy-25-00505-f009:**
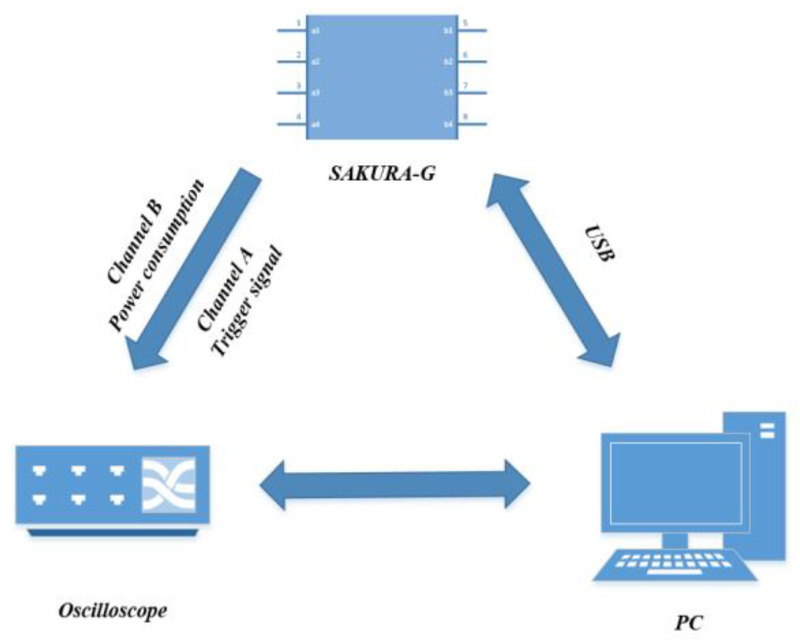
Experimental environment configuration diagram.

**Figure 10 entropy-25-00505-f010:**
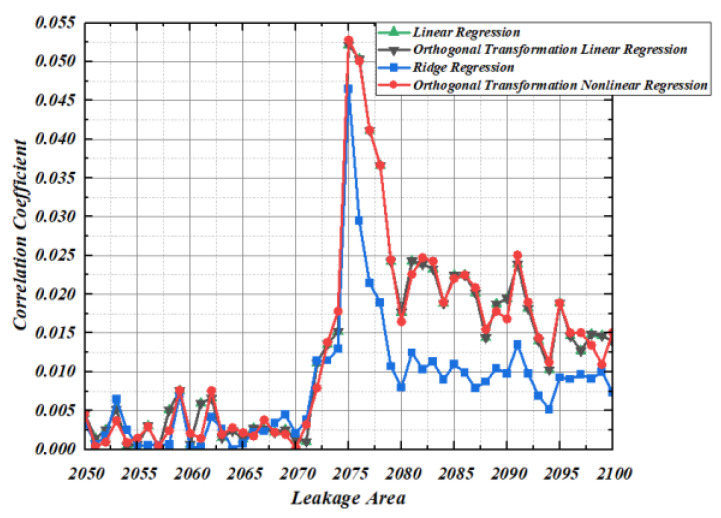
Correlation coefficient of Template.

**Figure 11 entropy-25-00505-f011:**
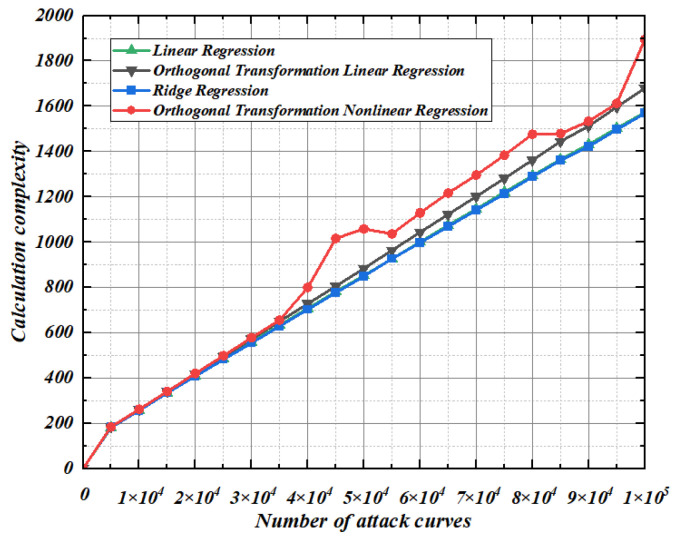
Calculation complexity.

**Figure 12 entropy-25-00505-f012:**
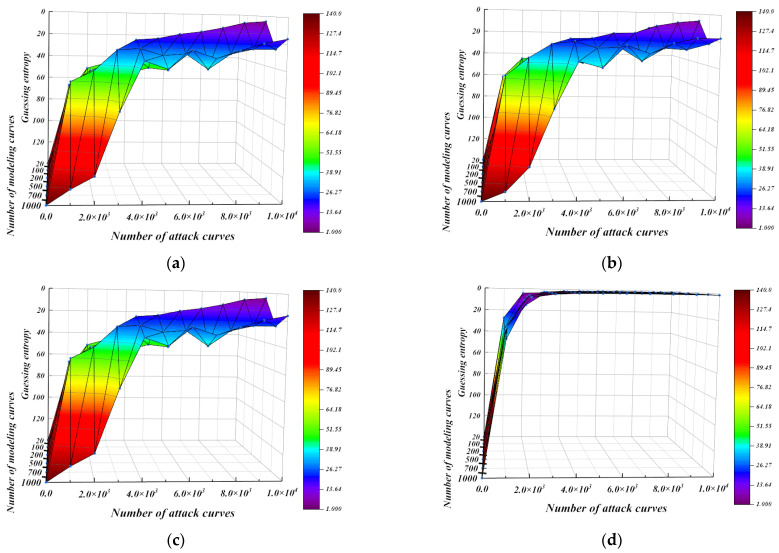
Guessing entropy under a different number of modeling curves and matching curves. (**a**) linear regression, (**b**) Ridge regression, (**c**) Orthogonal transformation linear regression, (**d**) Orthogonal transformation nonlinear regression.

**Figure 13 entropy-25-00505-f013:**
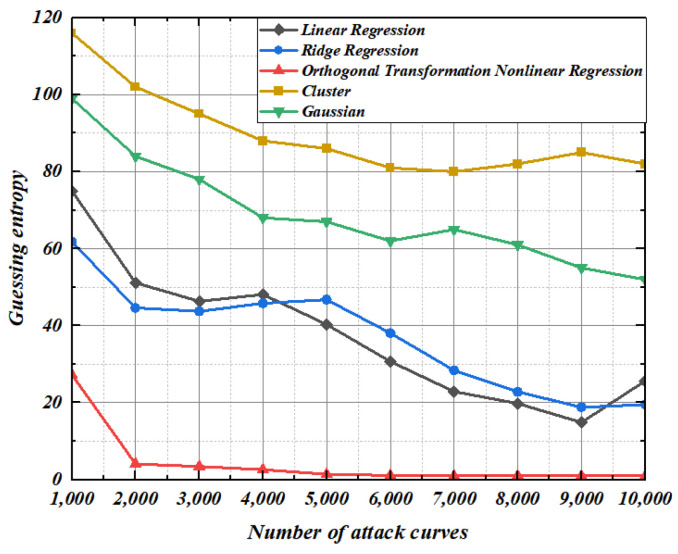
Construction: 2. Matching: 2.

**Figure 14 entropy-25-00505-f014:**
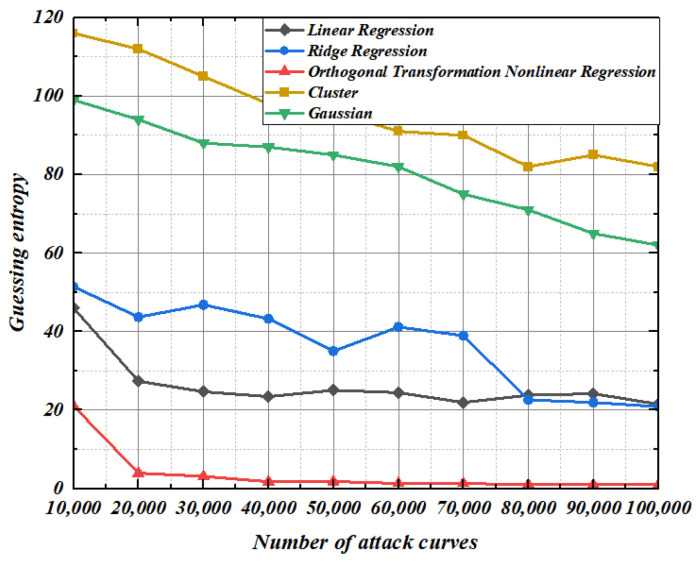
Construction: 2. Matching: 4.

**Figure 15 entropy-25-00505-f015:**
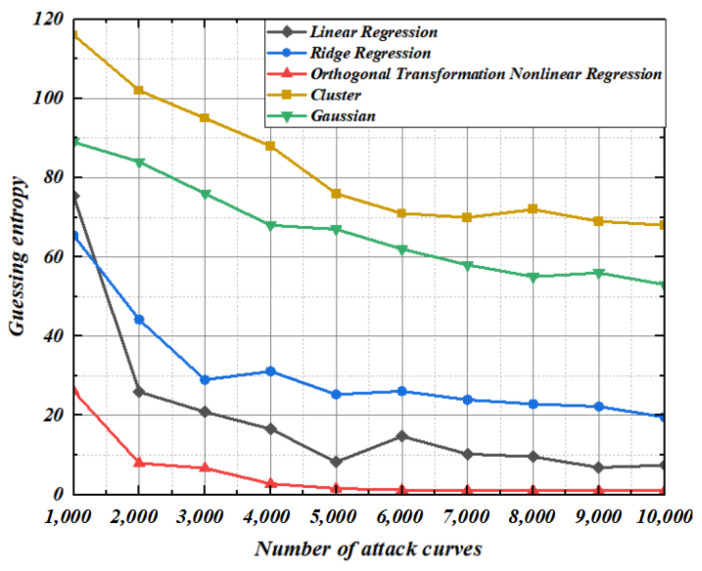
Construction: 4. Matching: 2.

**Figure 16 entropy-25-00505-f016:**
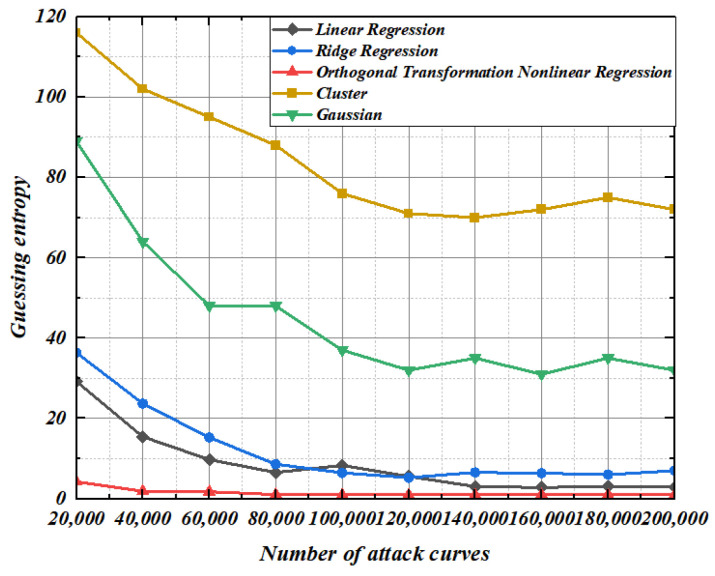
Construction: 4. Matching: 4.

**Figure 17 entropy-25-00505-f017:**
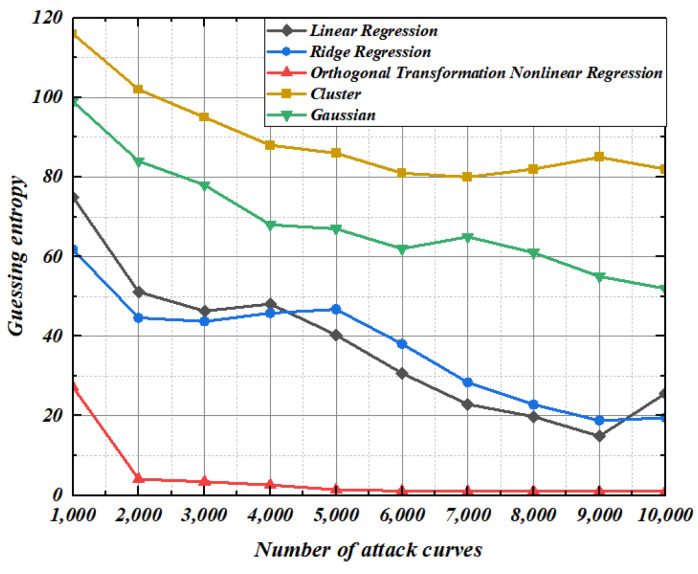
Second-order template analysis.

**Figure 18 entropy-25-00505-f018:**
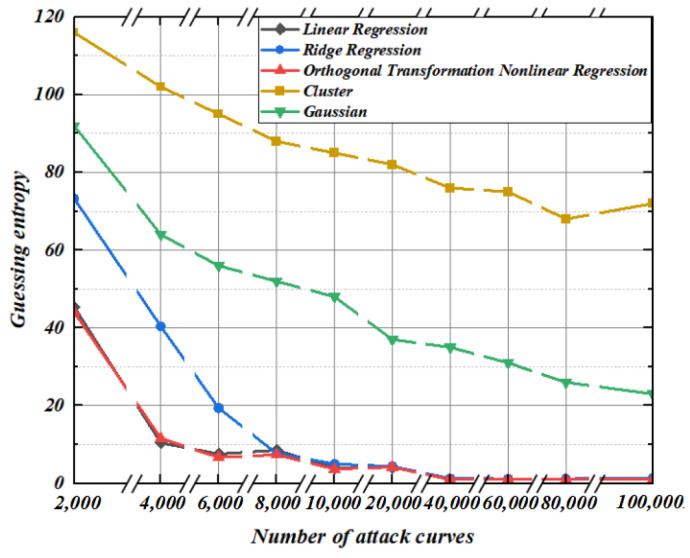
Fourth-order template analysis.

**Figure 19 entropy-25-00505-f019:**
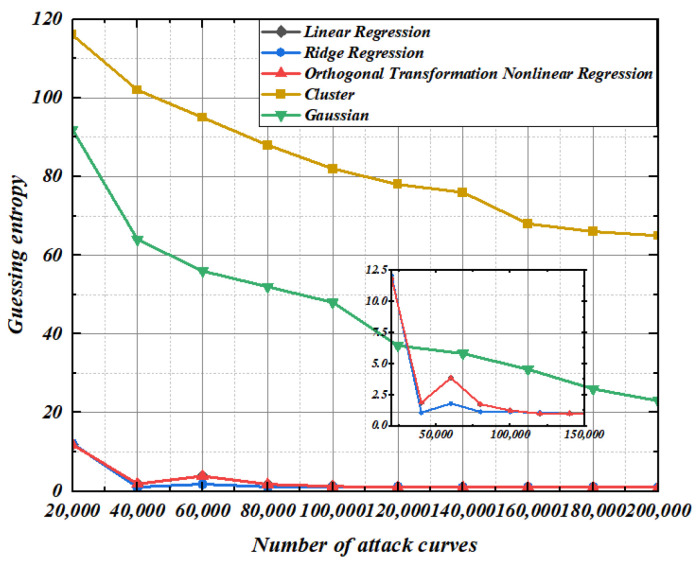
Eighth-order template analysis.

## Data Availability

All comparative data indicators can be found in the reference.

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
