# Peer review of "Second-Order Side-Channel Analysis Based on Orthogonal Transform Nonlinear Regression"

_entropy, 2023, doi:10.3390/e25030505_

Round 1
Reviewer 1 Report
My suggestions:
1. Introduction:
a) remove single characters ending lines,
b) unify citing range of resources [2 - 7] or [14-16] (no spaces).
c) unify the seperating distance: "fault analysis [19-21]" (with distance) or "regression[26-28]" (zero distance).
d) correct the usage of terminology, e.g. "summarize the thesis" (I think it's a paper).
3. Orthogonal Transformation Nonlinear Regression analysis:
a) Fig. 4 is having low quality.
5. Safety analysis
a) Is it "safety" or "security"?
b) Fig. 9, Fig. 10 are having low quality.
c) Figure 11(a) and (c) --> Figure 11(a) and 11(c)
6. Conclusion
a) Conclusion --> "Conclusions" or "Closing remarks"
b) Provide information about further works.
Author Response
- Introduction:
- a) remove single characters ending lines,
Removed according to the comments of reviewers.
- b) unify citing range of resources [2 - 7] or [14-16] (no spaces).
The format of references cited in the text is unified as [2-7].
- c) unify the seperating distance: "fault analysis [19-21]" (with distance) or "regression[26-28]" (zero distance).
The format of references cited in the text is unified as "regression [26-28]".
- d) correct the usage of terminology, e.g. "summarize the thesis" (I think it's a paper).
The author proofread and corrected the text again. For example, "Finally, we summarize the thesis in the sixth section." is changed to "Finally, we conclude this paper in Section VI.".
- Orthogonal Transformation Nonlinear Regression analysis:
- a) Fig. 4 is having low quality.
The author redraw Figure 4.
- Safety analysis
- a) Is it "safety" or "security"?
Correct "safety analysis" to "security analysis"
- b) Fig. 9, Fig. 10 are having low quality.
The author reprocesses Figures 9 and 10.
- c) Figure 11(a) and (c) --> Figure 11(a) and 11(c)
Correct "Figure 11 (a) and (c)" to "Figure 11 (a) and 11 (c)"
- Conclusion
- a) Conclusion --> "Conclusions" or "Closing remarks"
- b) Provide information about further works.
In the future, we may further study template analysis on this basis, and construct templates with more generalized key guessing efficiency and less key guessing cost by combining with clustering, binary classification and even neural networks.

Reviewer 2 Report
The paper is about template based side channel attack.
It might have some merit to it, but it was hard to read.
Critical:
The writing is too convoluted. It is not clear what the objective is.
There appear to have a lot of experimental result, but those are hard to make sense.
The authors may refer to https://dl.acm.org/doi/10.1145/3310273.3323399.
Editorial:
1. Abstract: "has become the biggest threat" -> one of the biggest
2. The figures look blurry, it is recommended to use Latex compatible tool (like, Tikz, Pgfplot, Matplotlib) and include as some vector graphics format (like PDF).
Author Response
The paper is about template based side channel attack.
It might have some merit to it, but it was hard to read.
Thank you very much for the suggestions of the reviewers. There are indeed many grammatical errors and writing mistakes in the paper. The author refinished the full text and re-established the structure of the paper.
Critical:
The writing is too convoluted. It is not clear what the objective is.
Template analysis mainly includes template construction and template matching. The construction of the template determines the success or failure of the attack. Among them, template analysis based on regression occupies a place in template analysis because it can construct efficient templates with few template curves. However, there are two problems in the template analysis based on linear regression and ridge regression. First, there may be irreversibility problems caused by singular matrices in linear regression analysis. Secondly, in the second-order or higher-order template analysis, because the ridge regression coefficient occupies too much space contributed by the original feature matrix, it may not be able to accurately fit the data. The experimental results show that its effect is even worse than linear regression. To solve the above problems, this paper proposes a second-order template analysis method based on orthogonal transformation nonlinear regression model. Orthogonal transformation projects data from the original space to the new space. At this time, the linear correlation variable is transformed into linear uncorrelated variable, and the irreversibility problem caused by singular matrix in linear regression analysis is solved without sacrificing the performance of regression estimation. At the same time, on the basis of linear regression, a negative direction is added in the calculation of the regression coefficient matrix, so that the least squares estimation of the regression coefficient is closer to the actual data. The template constructed by this method has good versatility, and has obvious advantages over the existing template analysis methods based on regression in terms of key guessing efficiency under high noise and high order conditions.
There appear to have a lot of experimental result, but those are hard to make sense.
In order to verify whether the template constructed by the second-order template analysis method based on the orthogonal transformation nonlinear regression model has good universality. Section 5 of the paper verifies the generality(key guessing efficiency) of the template constructed by this method by collecting power traces on the FPGA development board. Section 5.1 verifies the matching degree of templates constructed by different regression methods through the correlation between estimated power consumption and actual power consumption. Section 5.2 calculates and compares the computational complexity of different regression methods to guess the key. Section 5.3 observes the key guessing efficiency of different regressions through the two dimensions of template construction and template matching. Section 5.4 verifies the key guessing efficiency of this method under different noise conditions. Section 5.5 verifies the key guessing efficiency of this method under different order conditions. The above comparative experiments are all to verify whether the template analysis method based on orthogonal transformation nonlinear regression model can adapt to various scenarios to prove the universality of its template.
The authors may refer to https://dl.acm.org/doi/10.1145/3310273.3323399.
The author refines the full text and re-establishes the structure of the paper according to the references recommended by experts.
Editorial:
- Abstract: "has become the biggest threat" -> one of the biggest.
- The figures look blurry, it is recommended to use Latex compatible tool (like, Tikz, Pgfplot, Matplotlib) and include as some vector graphics format (like PDF).
Use origin and Visio to reprocess the blurred images in the text, for example, Figures 4, 9, 10,11.

Round 2
Reviewer 2 Report
It appears that the authors have improved the write-up. I would recommend to make the experimental data and source-codes public. The diagrams look hazy, it would be useful to see those in high resolution.